# Performance of early warning signals for disease re-emergence: A case study on COVID-19 data

**Daniele Proverbio** [1,2]*, **Françoise Kemp** [1], **Stefano Magni** [1], **Jorge Gonçalves** [1,3]

**1** Luxembourg Centre for Systems Biomedicine, University of Luxembourg, Belvaux, Luxembourg, **2** College of Engineering, Mathematics and Physical Sciences, University of Exeter, Exeter, United Kingdom, **3** Department of Plant Sciences, University of Cambridge, Cambridge, United Kingdom

* daniele.proverbio@uni.lu

**Data Availability Statement:** The analysis was performed in Matlab and Python. They require the Statistics Toolbox and the scipy library, respectively. Code and curated data are accessible

## Abstract

Developing measures for rapid and early detection of disease re-emergence is important to perform science-based risk assessment of epidemic threats. In the past few years, several early warning signals (EWS) from complex systems theory have been introduced to detect impending critical transitions and extend the set of indicators. However, it is still debated whether they are generically applicable or potentially sensitive to some dynamical characteristics such as system noise and rates of approach to critical parameter values. Moreover, testing on empirical data has, so far, been limited. Hence, verifying EWS performance remains a challenge. In this study, we tackle this question by analyzing the performance of common EWS, such as increasing variance and autocorrelation, in detecting the emergence of COVID-19 outbreaks in various countries. Our work illustrates that these EWS might be successful in detecting disease emergence when some basic assumptions are satisfied: a slow forcing through the transitions and not-fat-tailed noise. In uncertain cases, we observe that noise properties or commensurable time scales may obscure the expected early warning signals. Overall, our results suggest that EWS can be useful for active monitoring of epidemic dynamics, but that their performance is sensitive to certain features of the underlying dynamics. Our findings thus pave a connection between theoretical and empirical studies, constituting a further step towards the application of EWS indicators for informing public health policies.

## Author summary

To extend the toolkit of alerting indicators against the emergence of infectious diseases, recent studies have suggested the use of generic early warning signals (EWS) from the theory of dynamical systems. Although extensively investigated theoretically, their empirical performance has still not been fully assessed. We contribute to it by considering the emergence of subsequent waves of COVID-19 in several countries. We show that, when some basic assumptions are met, EWS could be useful against new outbreaks, but they may fail to detect rapid or noisy shifts in epidemic dynamics. Hence, we discuss the potential and

on: https://github.com/daniele-proverbio/EWS_epidemic.

**Funding:** DP's and SM's work is supported by the FNR (https://www.fnr.lu/) PRIDE DTU CriTiCS, ref 10907093. FK's work is supported by the FNR (https://www.fnr.lu/) fund PRIDE17/12244779/PARK-QC. The funders had no role in study design, data collection and analysis, decision to publish, or preparation of the manuscript.

**Competing interests:** The authors have declared that no competing interests exist.

limitations of such indicators, depending on country-specific dynamical characteristics and on data collection strategies.

## Introduction

Epidemics such as the current COVID-19 pandemic pose important and long-lasting threats to human societies [1]. Hence, developing tools for rapid and early detection of disease emergence is important to perform science-based risk assessment [2]. In principle, detailed mechanistic understanding could help formulate predictive models. However, combinations of non-linearity, noise and a lack of curated data sets hamper the development of mechanistic models. Therefore, numerous recent studies have suggested using different methods, agnostic of detailed mechanistic models, that could detect shifts in epidemic dynamics [3–7]. These methods are based on the theory of critical transitions in dynamical systems [8] and require the calculation of statistical early warning signals (EWS) from observed data. However, the applicability of such early warning signals is still debated, as it might depend on the interplay of modelling predictions and empirical observed dynamics.

Critical transitions encompass a broad class of complex phenomena characterized by sudden shifts in the system dynamics. Key mechanisms for deterministic shifts are dynamical bifurcations [9], i.e. qualitative changes of equilibria due to leading eigenvalues crossing a threshold value. In epidemiology, the leading parameter is the reproduction number $R$, the average number of secondary infections from a single contagious case in a susceptible population [10]. When this evolves over time, e.g. reflecting the effect of pharmaceutical or non-pharmaceutical interventions, we speak of effective time-dependent reproduction number $R(t)$ [11]. Re-emergence of infectious diseases thus involves a transmission system that is pushed over the critical point $R(t) = 1$ through a transcritical bifurcation [7]. As a consequence, it may in principle be possible to apply results from the theory of critical transitions to detect impending epidemic re-emergence. In particular, proposed early warning signals (EWS) are summary statistics indicators that might change in a predictable way when approaching the critical threshold. Common EWS are increasing variance and autocorrelation, which have been suggested to be generically applicable to detect impending regime shifts in different systems [12, 13]. If this would be the case, the consequence would be the possibility to expand the set of epidemic indicators. Several theoretical and computational studies already investigated EWS performance on abstract epidemiological models [4, 7, 14–16], but so far only a few testings on empirical data have been performed. A first observation was reported in [17]; further data-driven approaches have been applied in [18]. An approach derived from bifurcation theory on networks was applied on COVID-19 data in [19]. A review can be found in [20]. Performing more tests is thus a necessary next step towards the application of EWS in routine surveillance procedures. In addition, it is essential to characterise the potential confounders that might affect the expected signals.

In this study, we aim at testing the performance of EWS in detecting the re-emergence of observed epidemics, and at interpreting the observed performances based on the correspondence of modelling assumptions and dynamic features observed in the data. Specifically, we are not screening all possible EWS on all possible empirical data as it was done in [21, 22]; on the opposite, we are testing whether some EWS work when they are expected to, what happens in other cases, and why. In fact, theoretical predictions like early warning signals should ideally be tested in controlled experiments [23], but these are often not feasible in complex phenomena like epidemics. Instead, the present work considers curated observational data from many

outbreaks of the same disease, following the strategy of "natural experiments" [24]: first, constructing a data set that includes relevant time series data; second, accounting for possible confounders, i.e. dynamical characteristics that might alter the expected signals; third, evaluating the performance of EWS and interpreting it in light of previous theoretical results.

To this end, we use worldwide data from the current COVID-19 epidemic. The COVID-19 disease, caused by severe acute respiratory syndrome coronavirus 2 (SARS-CoV-2) [25], rapidly diffused in the whole world during 2020 and 2021. After a first outbreak, characterised by a sudden emergence followed by an exponential diffusion, countries worldwide managed to curb the local infection curves with combinations of non-pharmaceutical interventions. From a dynamical perspective, this corresponded to tuning the effective time-dependent reproduction number $R(t)$ to values below 1 [26]. Later during the year 2020, many countries experienced epidemic re-emergences (often called "second waves") associated to $R(t)$ re-crossing 1 from below [27, 28]. We thus concentrated on this re-emergence, in order to study signals associated with a bifurcation being crossed. The unprecedented diffusion of the virus, as well as the various degrees of intervention strengths, provide abundant epidemic data to construct the test set. Known dynamical features associated to modelling assumptions, such as noise and rate of evolution of $R(t)$ [29], are then accounted for by analysing the time series of each country. These features allow to interpret the trends observed in empirically derived EWS and their performance in different contexts.

The paper is organised as follows. First, we recall theoretical results from literature, to allow the subsequent comparison with expected EWS behavior. Then, we describe how the test set was constructed and analysed. After that, we study the behavior and the performance of EWS from empirical data and their dependence on dynamical characteristics associated with modelling assumptions. Finally, we discuss the current findings, their limitations and their implications for future studies.

## Methods and mathematical theory

### Mathematical theory and derivation of EWS

The scope of this article is to test theoretical predictions in light of their assumptions. Hence, we provide a brief review of the theoretical basis of critical transitions in epidemic dynamics, as well as on the derivation and assumptions that underlie their associated early warning signals. This way, we highlight the theoretical results to be tested as well as their supporting hypothesis, which will be central for this study. Further details can be found in supplementary S1 Text as well as in the references provided.

It is often recognised that epidemics can be described as complex systems, whose macro-scale dynamics evolves out of equilibrium [3, 7, 13]. In different complex systems, sudden dynamical changes can happen when the system is pushed over a critical point through a bifurcation [30]. Critical transitions observed in complex systems are often associated with such bifurcations [31]. Recent studies have shown that the trend of certain statistical indicators may signal the approach to a critical transition in slowly forced dynamical systems [9, 32]. In general, a slowly forced system with variables **x** and control parameter $q$ is [33]:

$$\begin{cases} \dot{\mathbf{x}} = \mathbf{f}(\mathbf{x}, q) \\ \dot{q} = \epsilon \mathrm{g}(\mathbf{x}, q) \end{cases} \tag{1}$$

with $0 < \epsilon \ll 1$. Typical models of epidemic dynamics, such as SIR based models [26, 34], can be expressed as a slowly forced system like Eq 1 when the control parameter $R$ slowly approaches its critical value 1.

In the presence of small random fluctuations, the approach to the critical point is often associated with predictable trends of several statistical indicators of variability, which have been proposed as early warning signals of impending transitions [35, 36]. The reason is that noise can push the system state around the deterministic trend of Eq 1; at the same time, its statistical properties might change as the system approaches the transition and could be used to detect it [33]. If the noise is relatively small with respect to the deterministic trend and normally distributed, the trend of the most common summary statistics (variance, autocorrelation, skewness, coefficient of variation), computed on detrended residuals, is expected to increase next to the transition, thus providing an early warning signal [13, 37]. If this signal can be observed prior to the transition, it would constitute an early warning.

Several early warning signals from the critical transitions theory have been predicted to apply on epidemiological models. A particularly relevant theoretical result is from [7]. The authors consider an extended SIR model:

$$\dot{X} = \mu(1 - p) - \beta XY - (\eta + \mu)X$$
$$\dot{Y} = \beta XY + \eta X - (\gamma + \mu)Y \qquad (2)$$
$$\dot{Z} = \mu p + \gamma Y - \mu Z$$

with variable $X$ for susceptible, $Y$ for infectious, $Z$ for removed. $\beta$ represents the infection rate and $\gamma$ the removal rate [38]; $\mu$ describes the flux of people across country boundaries; $\eta$ is an influx of infected cases that could trigger a new infection; $p$ represents a protection rate for the susceptible population, either by non-pharmaceutical interventions or by vaccination [39]. In this case, the reproduction number is [7]:

$$R = \frac{\beta}{\gamma + \mu}(1 - p). \qquad (3)$$

There is a transcritical bifurcation on the $(Y, p)$ diagram when $R$ reaches its critical value $R = 1$ for $p^* = 1 - (\gamma + \mu)/\beta$. The $R$ introduced above corresponds to the empirical effective reproduction number $R(t)$ if we explicitly consider the time-dependence of $p$ as $p(t)$. When considering the stochastic version of Eq 2, it is possible to analyse its fluctuations next to the transition and extrapolate the early warning signals from the associated summary statistic indicators. The kind of noise (additive or multiplicative) that is considered constitutes a modelling assumption. How the most common indicators—variance and lag-1 autocorrelation—are derived (from [7]) is reported in the supporting information, along with their predicted evolution next to the critical transition when the slow-fast assumption is satisfied, or not (Fig A in S1 Text). Their increasing trends prior to the transition provide the predicted early warning signal.

Further computational results, which also consider additional indicators, suggest that the increases in variance are the best performing indicators of re-emergence, in terms of signal-to-noise ratio and of detection performance [3, 4]. However, as noticed in follow-up studies [33, 40, 41], their performance is linked to whether their modelling assumptions are satisfied. If it is the case, such indicators perform well; but what happens in other contexts is still less clear.

We here recall the main modelling assumptions underlying the prediction of EWS, as well as their relevance for their performance. (1) Critical transitions are local phenomena. Hence, EWS are not global measures, but are expected to work in the vicinity of the regime shift. (2) It should be possible to express the epidemic dynamics in terms of a fast-slow system like Eq 1. When approximating $R$ approaching to 1 as a linear trend, the modelling assumption Eq 1 is satisfied if the regression coefficient (the slope of the linear trend) is small. Otherwise,

literature results suggest that the expected patterns will be either distorted or will not occur [3, 7]. (3) The closer random fluctuations are to be additive noise, the more robust the performance of EWS is. If there are deviations from white noise, EWS trends can be modified or disrupted. For instance, decreasing variance was observed in case of non-white multiplicative noise [41]. (4) In case of combinations of non-white noise and of non-fast-slow description, one might observe bifurcation delays, i.e. changes of the system state (and of its indicators) that lag behind the theoretical bifurcation. This would translate in a warning signal that emerges much time later than the epidemic re-emergence. (5) If the transition is triggered by large random fluctuations—the so called noise-induced transitions [9]—no EWS is expected to be observed [42].

## Data collection and curation

This paper studies the re-emergence of infectious diseases, with COVID-19 as a case study, in a number of observations from all over the world. Our aim is to verify whether EWS work when they are expected to based on the theory recalled above and explain why they might malfunction otherwise, rather than perform an observational study over all COVID-19 re-emergencies (for such a study, refer to [21]). Consequently, to construct the data set, we considered data from countries that faced a re-emergence of positive COVID-19 cases between beginning of March (starting of wide viral diffusion) to mid-September 2020. We did not consider further data points as many countries began issuing new social measures that rapidly impacted the epidemic trends. These would hinder the careful analysis of confounders.

When possible, we use prevalence data, i.e. active cases over the whole population of a regional area, in accordance to what is modeled by SIR-like models and to what was suggested in literature [14]. Active cases from Luxembourg are directly retrieved from the government website (COVID19.public.lu/fr/graph). They are derived from random samples over the whole population, using a Large Scale Testing strategy [43] and careful control of the hospital system. As they are not directly available for the other countries, active cases $A$ are estimated, following [44], by the proxy:

$$A = C - D - \tilde{R} \tag{4}$$

where $C$ indicates the cumulative positive cases, $D$ the number of registered deaths and $\tilde{R}$ the number of recovered patients. Country data are obtained from public repositories of confirmed detected, deceased and recovered cases: the John Hopkins University collection [45] and the European Centre for Disease Prevention and Control database (https://www.ecdc.europa.eu/en/COVID-19/data). We also use Italian data from the Veneto region, as an example of regional data with an identifiable second wave during the considered time interval. Veneto time series for detected, deceased and recovered cases are retrieved from the Github repository of the Italian "Dipartimento della Protezione Civile—Emergenza Coronavirus" (https://github.com/pcm-dpc/COVID-19). All databases are accessed up to 15/09/2020.

To best curate the database, an initial screening on data quality is performed. We reject time series with very few active cases, as in such time-series the intrinsic stochasticity of the contagions and the measurement noise dominate over the deterministic behaviour captured by SIR-like models. We also discard time series for which the share of positive cases over performed tests is $> 5\%$ next to the transition, as WHO guidelines suggest possible undertesting (we refer to WHO reports such as https://bit.ly/3dARcy1). Information about the share of positive tests is obtained from the OurWorldInData curated dashboard [46] and is reported as a summary in Table A in S1 Text. As EWS from critical transitions are based on mean-field homogeneous SIR-like models, we do not consider whole countries with clear spatial

heterogeneity like Italy [47], but we instead use regional data if available. Finally, we discard some public time series that behave clearly differently from the common reconstructed epidemiological curves [44] (see Fig B in S1 Text).

We acknowledge that data quality, particularly about $\hat{R}$ cases, plays a major role to obtain a robust estimator for $A$. The selection criteria were designed to enhance data quality; hence, in the remainder of this study, we focus on prevalence data to compare the results with interpretations from various literature sources. In addition, we perform an investigation on the use of incidence data, using reported daily new cases from the same sources listed above. Incidence data might as well be influenced by testing bias (e.g. lower testing over weekends) and other factors; hence, this analysis complements the one on prevalence data by investigating EWS performance on real-world monitoring protocols. Such analysis is reported in Sec H in S1 Text.

## Analysis of dynamical features

To identify the transition *a posteriori* and get a "ground truth" date of re-emergence, we use a data-driven estimation of the time-dependent $R(t)$. Similar to [4, 48, 49], $R(t)$ is estimated with Bayesian inference by means of a Markov Chain Monte Carlo (MCMC) method. For each day when data are available, we estimate the probability of observing a certain value of $R(t)$ by calculating the likelihood of seeing $k$ new cases, given the candidate $R(t)$, following a Poisson transmission process. To avoid fitting spurious bumps, the data are previously smoothed with a Gaussian window of 7 days. Note that, since this is only used for a retrospective analysis, it does not modify the non-anticipating scheme for the EWS. We update the prior at time $t$ with the posterior at time $t - 1$. A Metropolis-Hastings MCMC scheme was used to generate candidates for $R(t)$. We describe this in depth in Sec D in S1 Text. As we adapted a previous implementation from [48], we also refer to it for further details.

Then, we employ the posterior probability density function obtained from the Bayesian framework: $p(R|\text{data})$. This was used to estimate the probability that the control parameter is greater than 1, $\mathcal{P}(R(t) > 1)$. This was calculated, as for any stochastic variable, by integrating over all possible probability values associated with $R(t) > 1$:

$$\mathcal{P}(R(t) > 1) = \int_1^\infty p(R|\text{data})dR \ . \tag{5}$$

Since $R(t) > 1$ is associated with an exponential increase of infectious cases after a transcritial bifurcation, $\mathcal{P}(R(t) > 1)$ can be interpreted as the probability of seeing an epidemic outbreak. Then, the most likely day $t_{em}$ in which the transition happened, assumed as our ground truth, corresponds to the first time when $\mathcal{P}(R(t) > 1)$ from Eq 5 reaches its maximum value of 1 (see Fig C in S1 Text).

After calculating the outbreak date, we test the modelling assumptions of normally distributed fluctuations and of slow approach to the critical transition.

To test the additive noise assumption, we analyze the global distribution of stochastic fluctuations, filtered from the time series with a 7-days moving Gaussian kernel as suggested in [50, 51]. The window size reflects typical cycles of data reporting and of COVID-19 fluctuations [52]. The distribution of fluctuations over the complete time series is indicative of the average noise distribution. We computed skewness and kurtosis to measure deviations from Gaussian noise, which is characterized by skewness = 0 and kurtosis = 3 [53].

To test the assumption of slow approach to the transition, we measure the rate of approach of the control parameter to its critical value. For this, we compute the time-dependent $R(t)$ like above and, consistently with the fast-slow system description of Eq 1, we fit a linear function:

$$R(t) = a + b \cdot t \tag{6}$$

in the interval $t \in [\tilde{t}, t_{em}]$. Here, $t_{em}$ corresponds to the day associated with novel disease emergence as explained above; $\tilde{t}$ is the day associated with the minimum of $R(t)$ after the first wave. The regression coefficient $b \pm \sigma_b$ measures the ramping speed of the control parameter (along with its uncertainty). As an indication, $R(t)$ is said to be "slowly evolving" if it goes from its minimum value to 1 in a period of time that is much longer than the COVID-19 serial interval (around 4 days [54]), which is a proxy of the disease duration time scale. For the fitting, we use the *scipy* Python library. Refer to Sec E in S1 Text for details.

## Estimation of EWS

Estimation of early warning signals from time series data is performed following standard methods from literature [29].

First, we detrend the time series to obtain a moving average, representative of the deterministic trend. The "residuals" or detrended fluctuations are obtained by subtracting the moving average from the original time series. To investigate possible effects of detrending approaches—as discussed in previous theoretical studies [13, 32, 50]—we use and compare three detrending methods: a uniform moving mean, a Gaussian kernel, and ARIMA models [55]. The ARIMA models are specifically tuned for each country, see Table B in S1 Text and Table D in S1 Text.

Then, we compute the statistical indicators associated to each point with a backward sliding window, i.e. one where the associated time point is the rightmost one. In a similar spirit, all detrending methods are non-anticipating. This way, all estimates are agnostic of future values and reflect practices used in active monitoring: the estimation of an indicator is performed as soon as a new data point becomes available. All EWS indicators are estimated on the detrended time series. We initially calculate the variance, which is suggested to be the most robust indicator for epidemic re-emergence [3, 4, 7, 14] as:

$$\text{Var}_{i,t} = \frac{1}{M-1} \sum_{s=t_0}^{t} (A_{i,s} - \hat{A}_{i,s})^2 \tag{7}$$

for any time point $i$ with active cases $A$, over a sliding window with size $t - t_0$ including $M$ time points. $\hat{A}$ is the moving average. We also estimate other common statistics such as lag-1 autocorrelation AC(1), coefficient of variation (CV) and skewness, which are constructed similarly to the variance over the same sliding window. The sampling frequency of COVID-19 data is not sufficient to allow estimation of the power spectrum reddening [56] or of the sample entropy [3]. All indicators are estimated with their corresponding MATLAB functions. Note that the estimation of $\mathcal{P}(R(t) > 1)$ is done a posteriori, that is, once we know the complete time series. Instead, the early warning signals are calculated a priori, without knowing in principle if a transition is approaching.

## Quantification of EWS trends and receiver operator characteristics analysis

Recent studies [14, 29, 40] suggest to quantify the expected increasing trend of EWS next to the transition with the Kendall's $\tau$ coefficient of monotonicity. The Kendall's $\tau$ score is defined as [57]:

$$\tau = \frac{\# \text{ concordant pairs} - \# \text{ discordant pairs}}{M(M-1)/2}.$$

$M$ is the number of considered time points. Two generic points $(t_1, x_1)$ and $(t_2, x_2)$ are said to be a concordant pair if, for $t_1 < t_2$, $x_1 < x_2$, and a discordant pair otherwise. A constant trend is expected to have $\tau = 0$. We compare this value with the $\tau$ scores calculated on time

series with identified transitions. To go beyond simple visual inspection, we quantify the detection power of each statistical indicator using its time-changing trend, classifying data as either belonging to the second wave or not. After calculating each indicator on a moving window (its size is discussed later in the text) for each detrended time series, we estimated the Kendall's $\tau$ score for each timepoint on windows of the same size, over an overall period $-30 < t_{em} < 5$ days around the transition, as our positive data set. $t > -30$ is chosen to avoid significant overlaps with the first epidemic wave, $t < 5$ to account for possible small bifurcation delays [13]. For the negative data set, we use $\tau$ values taken way before the transition occurs, that is on windows associated with timepoints $t < -30$.

We use Receiver Operator Characteristics (ROC) analysis to classify each time point as either before or after re-emergence. We compare each statistical indicator's ability to correctly distinguish which Kendall's $\tau$ scores belong to those from before or after re-emergence, that is, we determine whether the estimated $\tau$ is higher or lower than some threshold value at that timepoint and determine whether each time series is classified correctly by that threshold. This gives a proportion of true positives and false positives. To do so, we compare various values for $0 < \tau < 1$ to those of the positive and negative data set, for each country. We calculate the indicator for each country in a test set at the given timepoint, and then group the specificity and sensitivity results to obtain the final ROC curve. The ROC analysis returns the Receiver Operator Characteristics (ROC) curve, a parametric plot of the sensitivity and specificity of a classification method as a function of the detection threshold [4, 58]. The overall detection performance of each EWS is quantified by the area under the ROC curve (AUC). A value AUC = 0.5 means that the statistics detection performance is as good in classifying as randomly guessing. A good indicator should have AUC close to 1, which informs that it is possible to identify the transition by the increasing trend of the indicator. An AUC close to 0 indicates good classification, although resulting from a decreasing indicator that does not correspond to the predetermined theoretical prediction.

## Results

### Analysis of country-wise dynamical characteristics associated to the spread of COVID-19

Table 1 reports the list of countries that satisfy the curation requirements discussed in "Methods and Mathematical Theory" and are thus included in the analysed data sets. Table 1 also reports the dates of re-emergence, identified by the analysis of $R(t)$. Fig 1A shows an example

**Table 1. Selected countries for the dataset, abbreviations and date of second epidemic insurgence.** Refer to "Data Collection and Curation" for how the date marking the second wave is obtained.

| Country | Abbr. | Date |
|---|---|---|
| State of Victoria (Australia) | AUS | 27/06/2020 |
| Austria | AUT | 01/07/2020 |
| Denmark | DNK | 03/08/2020 |
| Israel | ISR | 01/06/2020 |
| Japan | JPN | 28/06/2020 |
| Korea, South | KOR | 13/08/2020 |
| Luxembourg | LUX | 29/06/2020 |
| Nepal | NPL | 29/07/2020 |
| Singapore | SGP | 25/07/2020 |
| Veneto (Italy) | VEN | 29/07/2020 |

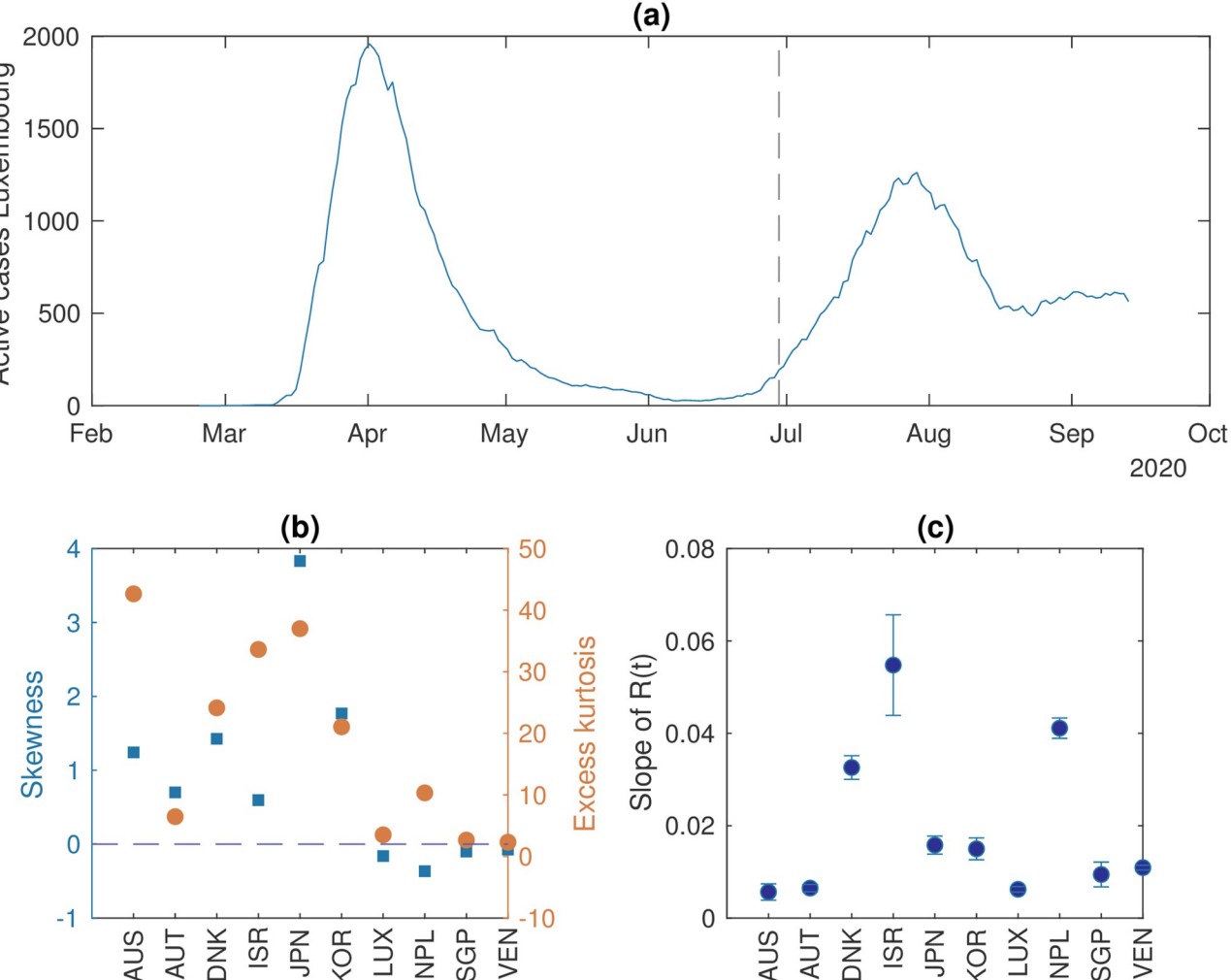

**Fig 1. Analysis of the dynamical characteristics of the countries included in the data set.** A: An example of an epidemiological curve of active cases from Luxembourg. The dashed line indicates the transition, measured by $R > 1$. The latter is objectively identified by the date at which the probability of $R(t)$ to be greater than 1 is at its first maximum (see "Analysis of dynamical features"). B: Measures of the distribution of data fluctuations. Skewness $\mu$ indicates the symmetry of the distribution, whereas kurtosis $\gamma$ indicates the relevance of its peak with respect to the tails. Large deviations from $\mu = 0$ (dashed line) and $\gamma = 3$ are associated with non-normal distributions, so we display the excess of kurtosis $\gamma - 3$. C: The regression coefficient of $R(t)$ and its associated uncertainty, as obtained from the linear fit Eq 6.

of time series of active cases for Luxembourg, from March to mid-September 2020, with the date of estimated re-emergence (dashed line).

The time series of the considered countries show different dynamical features. Fig 1B shows various ranges of noise distribution, measured by skewness and kurtosis. Austria, Luxembourg, Nepal, Singapore and Veneto display noise distributions that are close to Gaussian, while noise in the other countries is further away from white than in the previously mentioned ones. This could be associated to social dynamics or imperfect data reporting [59].

The rate of approach of $R(t)$ to its critical value also differs, as indicated in Fig 1C by the regression coefficient of a linear fit for $R(t)$ (*cf.* Eq 6). State of Victoria, Austria, Luxembourg, Singapore and Veneto display a slow approach to the critical value and can thus be better suited to be appropriately described as slow-fast systems like Eq 1. Japan and South Korea

show intermediate values, while other countries—Denmark, Israel and Nepal—have a faster evolution of the control parameter, which does not satisfy the assumption of slow evolution.

Following this analysis, we subdivide the considered countries into two test sets, based on whether the country satisfies or not some of the assumptions of slow approach to the transition and noise distribution close to white. These two properties can be assessed respectively from Fig 1C and 1B. Instead of using hard thresholds, we use group clustering to make the subdivision. As discussed above, State of Victoria, Austria, Luxembourg, Singapore and Veneto are grouped together to represent the assumption of slow rate, as can be seen from Fig 1C. Moreover, except for Australia, their noise distribution is close to Gaussian, as their skewness and kurtosis show in Fig 1B. They thus form the test set $\mathcal{Y}$, used to further assess the performance of EWS. On the other hand, Denmark, Israel and Nepal display higher rates of approach to $R(t) = 1$ and large deviations for Gaussian noise distribution. Hence, they are grouped together in a set $\mathcal{N}$; this is used to interpret the performance of EWS in settings that are not properly described by theoretical models and represent possible limitations of the predetermined predictions. South Korea and Japan are more ambiguous when clustering over the slope of $R(t)$, therefore we split them into $\mathcal{Y}$ and $\mathcal{N}$, respectively, based on their relative vicinity to Gaussian noise distribution.

Among these countries, Luxembourg is peculiar, as it satisfies the modelling assumptions and is the closest to being a "controlled experiment" according to the criteria described in the section about deriving EWS. In fact, we know from literature and practical experience that the country is small, homogeneous population-wide interventions were in place, and a Large Scale Testing (LST) strategy was implemented to best monitor the virus diffusion in the country [43]. This country wide testing strategy reached more than 70.000 tests per week over a population of about 600.000, thus allowing extensive and frequent monitoring. Hence, we use it as an initial sample to test the theoretical predictions about the local behavior of EWS.

## Local trends on controlled data and impact of detrending methods

We first focus on Luxembourg, that displays the best data in terms of curation of prevalence data (see "Methods and mathematical theory") and of satisfaction of theoretical assumptions (see above). Here we test the theoretical predictions about the local behavior of common EWS. Summary statistical indicators are estimated from the detrended fluctuations (residuals) around prevalence data as per standard methods [29].

We first investigate the effect of the detrending method in generating residuals. To do so, we compare the fluctuations around the deterministic trend obtained with a Gaussian kernel smoothing [32, 50], a moving average filtering [13] and an ARIMA(2,1,3) model [55]. Fig 2 shows the time evolution of the residuals obtained with the three methods, as well as their mutual correlations. As quantified by the Pearson correlation coefficient, the Gaussian and moving average filtering have similar output (correlation coefficient $\rho = 0.95$). This is likely related to the Gaussian bandwidth of 7 days, used to reflect known weekly fluctuations related to testing routines. Consequently, the Gaussian kernel smoothing is used in the rest of the analysis. However, the ARIMA method returns residuals that are less correlated with the previous ones ($\rho = 0.23$), whose effect on EWS needs further investigation.

Then, we study the behavior of the variance (theoretically, the most robust EWS [4]) next to the transition point, identified as the day when the estimated $R(t)$ crosses 1 (dashed line in Fig 3). The increase in variance prior to the transition, as expected from theoretical studies, is evident in Fig 3, irrespective of the moving window size and on the detrending method. Although the lead time is slightly advanced for shorter window sizes, the corresponding Kendall's $\tau$ measure of monotonous increase is similar for both methods and all window sizes (*cf.*

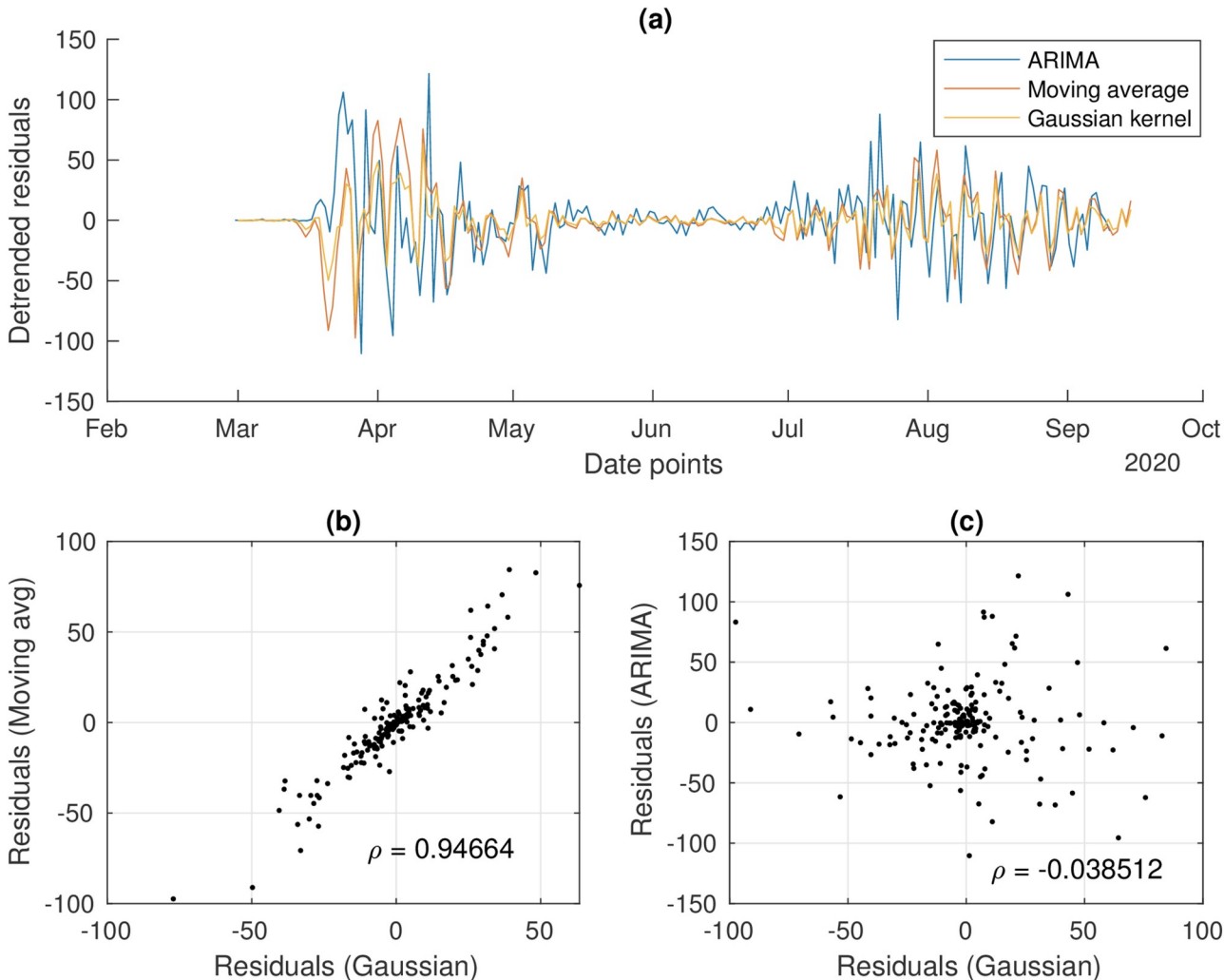

**Fig 2. Analysis of the residuals from the detrending methods (case study from Luxembourg shown).** a) The detrended fluctuations time series. b) Correlation between residuals obtained from Gaussian or moving average filtering. c) Correlation between residuals obtained from Gaussian or ARIMA filtering.

values reported in Fig 3). In general, a large window size produces smaller fluctuations but a visually reduced absolute increase; in addition, too large windows might capture old decreasing trends, that we want to avoid analysing. On the contrary a small window size is associated with less smoothed curves but a larger absolute value of variance. Nonetheless, it might not include enough data points to capture the trends in more noisy estimators like AC(1) [50]. From here on, we will use a window of 14 days as a reasonable trade-off, collecting enough data to be robust without being over-dependent on past history. The ARIMA residuals produce a visually clearer increase in variance, but the Kendall's $\tau$ quantifies an analogous trend (even slightly lower). For the incoming quantitative analysis on the EWS performance, we will thus study the effect of both detrending methods.

These findings confirm that, in a controlled setting that satisfy the modelling assumptions (Luxembourg), an increasing trend of the variance in the vicinity of the transition point could serve as early warning to detect the transition to disease emergence.

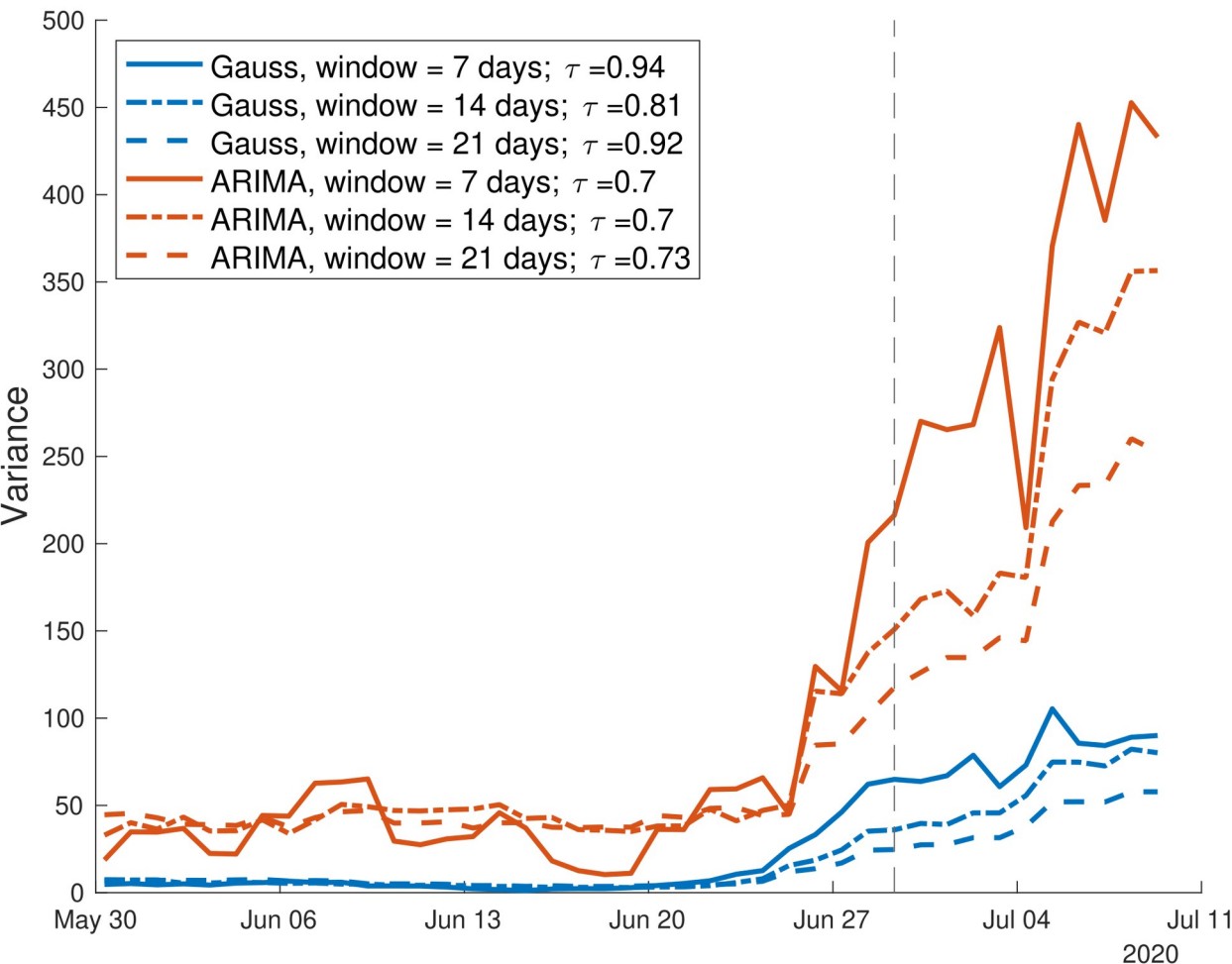

**Fig 3. Analysis of the variance in the Luxembourg setting.** Its increase is evident prior to the transition (dashed vertical line). $\tau$, which quantifies the overall increasing trend, is little sensitive to the sliding window size, as displayed by the three curves and by $\tau$ values reported in the text. The variance is computed over the residuals from Gaussian filtering and ARIMA detrending. The increasing trend during the considered time window is quantified by the associated $\tau$ values.

### Global trends of EWS

After confirming the local behavior of the variance in the Luxembourg highly controlled setting, we widen the analysis to the global performance of other EWS, *i.e.* far from the bifurcation, and for different countries from the pre-defined dataset, *cf.* Table 1. This way, we further test the theoretical predictions and the EWS potential use in more general contexts.

Among the indicators, we estimate lag-1 autocorrelation (AC(1)), skewness and coefficient of variation (CV), which are often proposed as alternatives to the variance. The size of the moving window is set to 14 days as discussed above. To compare the trend of EWS with the approach to the bifurcation, the probability of $R(t)$ to be greater than one (from Eq 5) is also calculated and reported.

Fig 4 shows the results for Luxembourg, Austria, State of Victoria (from the test set $\mathcal{Y}$). In addition, Israel, which does not satisfy the EWS assumptions (*cf.* Fig 1) is reported to inspect a deviant case. The figure focuses on EWS trends after the first wave, up to about a month after the second epidemic insurgence. The graphs for other countries from Table 1 are reported in

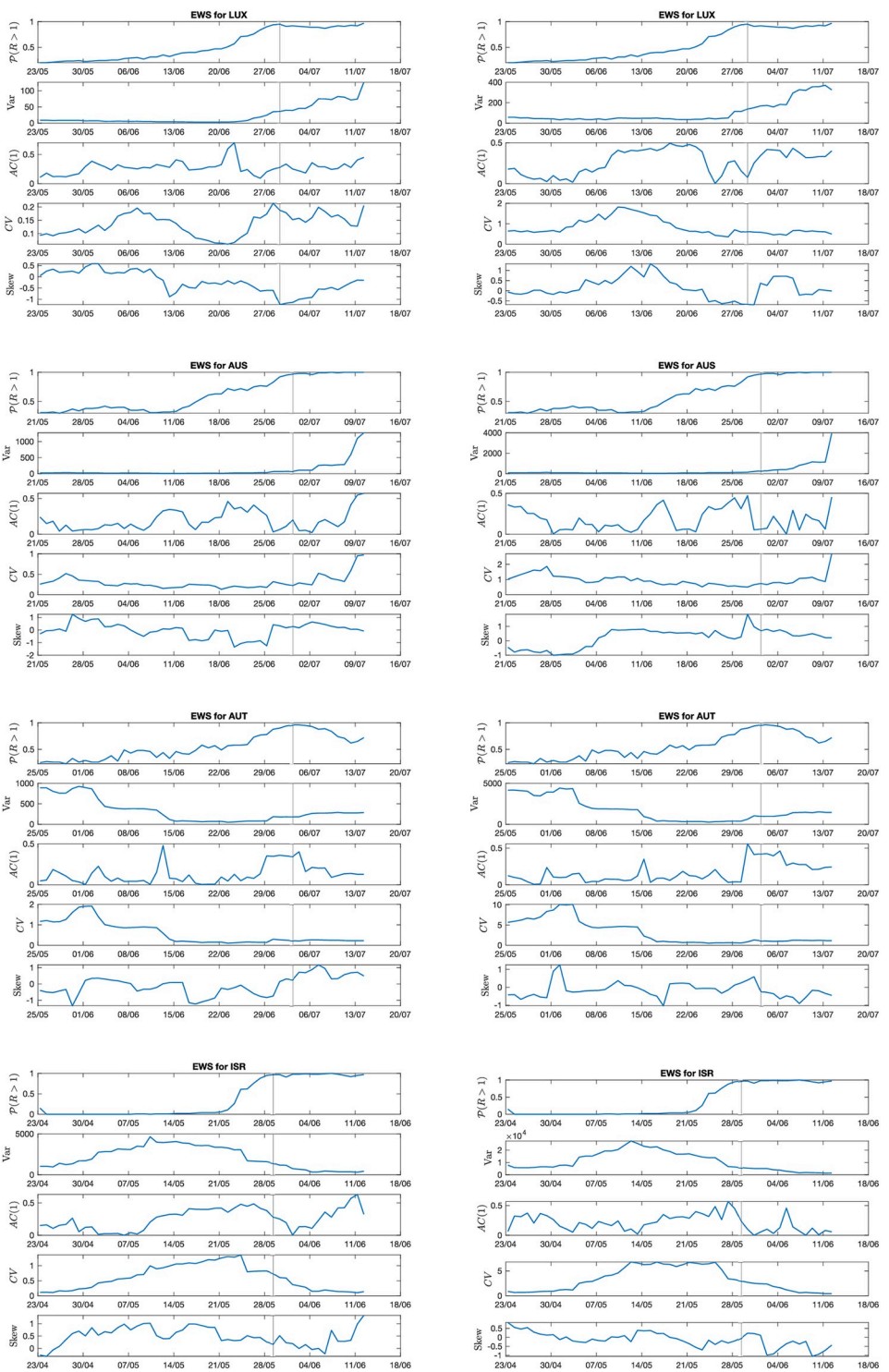

**Fig 4. Evolution of EWS far from the transition point.** Four example countries are shown: Luxembourg and Austria, with controlled features; State of Victoria (Australia), with small deviations from controlled features; and Israel that does not satisfy theoretical conditions. Considered EWS are the most common ones (variance, lag-1 autocorrelation, coefficient of variation, skewness). In addition, to mark the approach to the transition, $\mathcal{P}(R(t) > 1)$ from the Bayesian estimation (see Eq 5) is displayed. The vertical line reports the transition date. Left column: detrending method employed: Gaussian filtering. Right column: detrending method employed: ARIMA. Other countries are reported in S1 Text.

Fig E in S1 Text, along with their associated prevalence data and estimated $R(t)$. In Fig 4, the left column refers to indicators estimated after Gaussian filtering, while the right column is for indicators estimated after ARIMA detrending.

Focusing first on Luxembourg and Austria, the variance follows its theoretically predicted behavior closely (*cf.*, e.g., Fig 9b in [7] and Fig A in S1 Text), with a small but visible increase prior to the transition and a subsequent monotonous trend along the second wave. In Austria, though, it still displays some fluctuations after the relaxation of the first wave. The same happens for the coefficient of variation CV, which depends on the variance and on a stable equilibrium in infectious numbers. On the other hand, the lag-1 autocorrelation shows an increasing trend very close to the transition point, but gives possibly spurious signals during the global time series. Finally, the skewness does not display immediately detectable relevant trends, as anticipated by computational studies [14]. This might be related to noise properties, as suggested by [60].

Variance and CV on Australian data, when processed by eye, start increasing close to the transition, but this becomes more pronounced around the $7^{th}$ of July. This might be related to the so-called "bifurcation delay", which is associated with deviations from Gaussian noise [13, 41], or to delays due to tests results reporting or symptoms onset.

Finally, Israel provides an interesting case study as it diverges from the theoretical assumptions, see Fig 1. In fact, its transition to epidemic re-emergence is rapid, and the noise distribution is far from being Gaussian. These characteristics disrupt the EWS trends as predicted by the theory. In fact, the variance remains flat around the transition (it is even slightly decreasing), CV and skewness slightly decrease, while lag-1 autocorrelation does not display informative patterns. A delay is reported more than 20 days after the transition, but it is as abrupt as the exponential increase in infectious data (see electronic supporting information). This shows that the application of early warning signals indicators on appropriate contexts is crucial to obtain reliable signals for developing risk assessment analysis.

Similarly to what observed in Fig 3, the variance trends are similar between Gaussian- and ARIMA-related indicators. We quantify their potentially different performance in the next section. The behavior of CV and AC(1) is also qualitatively rather similar. On the other hand, the skewness behaves differently. For instance in Austria, it increases when the detrending is performed with Gaussian kernel, but it decreases after ARIMA. This might be associated to the fact that the skewness is very sensitive to the noise distribution [60]: small changes in the residuals, due to the different filtering procedure, might suffice to modify its trend.

## ROC quantitative analysis of EWS performance

For the online detection of incoming re-emergence, distinguishing between robust increases and spurious fluctuations is crucial to optimise the true positive signals and minimise the false negatives. Hence, a retrospective analysis of time data is often not sufficient and is only useful for offline detection. Thus, we provide a quantitative estimation of EWS performance in robustly detecting the transition. The Kendall's $\tau$ score is used to evaluate if a certain indicator corresponds to an increasing or decreasing trend and compare this for different data types [14, 40]. Hence, we evaluate $\tau$ for each indicator, over the same 14 days window, and we assess which values are associated with a passage through the transition point. The increase in $\tau$ is reflected in the Receiver Operator Characteristics (ROC) curve and quantified by Area Under the Curve (AUC) scores. Fig 5 shows the ROC curves for the considered indicators averaged over the countries in $\mathcal{Y}$. Panel (a) reports ROC curves calculated over data detrended with Gaussian filtering; panel (b) focuses on ARIMA detrended data. Table 2 reports the corresponding AUC values, for both methods. The variance is the only indicator that consistently

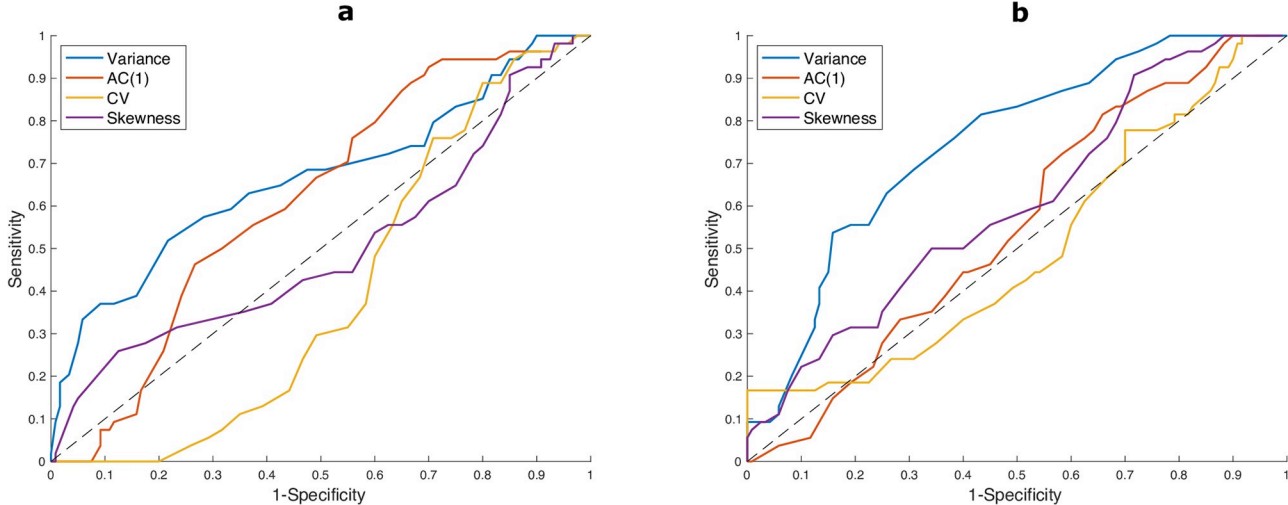

**Fig 5. ROC curves for each considered indicator, with sensitivity and specificity calculated on each timepoint for all countries in $\mathcal{Y}$.** Each point corresponds to a test value for $\tau$, to define if the detection is positive. The diagonal line corresponds to the ROC of a random classifier. Curves above it imply better performance. a) Computed on Gaussian filtered data; b) Computed on ARIMA detrended data.

performs better than a random classifier, while the lag-1 autocorrelation seldom performs slightly better than that. This is in line with aforementioned theoretical results from literature, e.g. [4, 14]. Instead, the skewness does not improve the detection performance. This is probably due to its fluctuations around the 0 value, as noticed in Fig 3, which is in turn associated with noise distribution of original data. Interestingly, the coefficient of variation is overall the worst performer. We speculate that this is due to its sensitivity to data fluctuations, which are often non negligible even in countries that belong to the test set $\mathcal{Y}$ (*cf.* Fig 1). We acknowledge that our findings are sensitive to the estimated time of emergence, which also complicates the estimation of the lead time. Currently, the best lead time is of 5 days for Luxembourg, a setting that is close to the analytical assumptions.

For all indicators, the ARIMA detrending method yields better performances, quantified by higher AUC values. AC(1) is an exception, as both methods return similar values, close to the ones of a random classifier. That ARIMA residuals yield higher AUC is potentially linked to the ARIMA estimating trends more closely at different time scales, thus returning more accurate fluctuations; on the contrary, the Gaussian filtering might be slightly more rough in considering only average time scales and returns approximated estimates for the fluctuations. This argument might explain why the skewness performs slightly better than a random classifier over ARIMA residuals: by considering more fine-grained time scales, the ARIMA seems able to pick the slight asymmetry in residual distributions that yield skewness-related signals [60].

**Table 2. AUC scores for different indicators, over $\mathcal{Y}$ and $\mathcal{N}$ datasets, after Gaussian or ARIMA detrending methods.**

| Indicator | Gaussian det. | | ARIMA det. | |
|---|---|---|---|---|
| | **over $\mathcal{Y}$** | **over $\mathcal{N}$** | **over $\mathcal{Y}$** | **over $\mathcal{N}$** |
| Variance | 0.6671 | 0.1981 | 0.7123 | 0.0934 |
| AC(1) | 0.5258 | 0.4995 | 0.5182 | 0.2840 |
| CV | 0.3968 | 0.1043 | 0.3626 | 0.0368 |
| Skewness | 0.4664 | 0.2925 | 0.5482 | 0.4609 |

This analysis thus indicates the importance of choosing the detrending method to increase the detection performance of various indicators.

The same analysis was performed over the set $\mathcal{N}$ of countries that do not satisfy the theoretical assumptions. Their AUC values are reported in Table 2. Such values clearly show that the considered indicators are not able to detect the transitions, overall performing worse than random classifiers. This supports what was already noticed for Israel in Fig 4, where disrupted trends were observed, contradicting what was expected and thus returning a false negative signal. For variance and CV, an AUC value close to 0 indicates that the transition is well detected by decreasing trends. This would contradict the theoretical predictions. Investigating this issue reveals that such features are possibly linked to non-complete relaxation of the indicators after the first wave or to delays. We thus conclude that it is an instance of spurious signal, to be carefully interpreted. See also Sec G in S1 Text and Fig F in S1 Text for further discussion. The time series for indicators of other countries in $\mathcal{N}$ are also reported in Fig E in S1 Text. Hence, if a system is not known or there is difficulty in determining the type of data, incorrect conclusions could be drawn when interpreting the time series trend.

## Discussion

Research on early warning signals from the theory of dynamical systems has greatly progressed in the last years, with a relevant focus on disease re-emergence. However, verifying and interpreting empirical analysis according to theoretical assumptions has been so far limited. In this study, we observe that some EWS from the critical transitions framework are able to detect the transition to disease re-emergence when necessary theoretical assumptions like normal distribution of data fluctuations and slow change rates are satisfied. On the contrary, we observe (along with [21]) that noise and commensurable time scales can obscure the early warning signals, which calls for caution when interpreting monitoring outputs. We suggest that dynamical-based EWS can be suitable candidates for epidemic monitoring. Theory-based indicators can provide useful evidence, particularly when scarce data and few prior information are a constraint for using large scale statistics or Machine Learning (ML) methods. However, their alternate performance on unknown dynamics asks for careful assessment of the underlying dynamics. EWS have the potential to complement the existing toolbox of indicators to improve epidemic risk assessment and deserve further investigation by scholars and decision-makers.

To support the growing corpus of theoretical studies, this study tested whether observed epidemic outbreaks behave consistently with the theory. When randomised experimental studies are not possible, observational studies provide stronger evidence if consistent patterns are seen in multiple locations and at multiple times, after checking for possible confounders. Hence, we employed world-wide available data about the ongoing COVID-19 pandemic, concentrating on the re-emergence of the disease after a first wave in spring 2020. To limit biases associated with country-specific testing and reporting capacities, we constructed a limited, but curated set of time series data. Like $R(t)$ and other indicators [61], EWS are estimates that rest on assumptions; hence, we screened the dataset to assess the matching of empirical features and theoretical assumptions. This pre-analysis showed that the same disease in diverse countries might have different noise distributions and evolve at different rates, which could depend on several factors including human behavior and population-wide interventions [62].

Furthermore, we tested how the best performing EWS detected the epidemic re-emergence. We carried out an extensive analysis of the effect of different detrending methods, showing that they are overall robust in highlighting the local trends of EWS. Such trends were studied both qualitatively and quantitatively, with ROC/AUC values. In particular, the ROC analysis

assesses the robustness of online detection in distinguishing between real increasing trends and spurious fluctuations, which are often not studied in retrospective observations of time series data. In controlled settings, our results reconstructed the expected trends in early warning signals and the potentials of the indicator system proposed. In particular, we showed that dynamical EWS are likely to operate successfully in contexts where the approach to the transition is gradual and not subject to high fluctuations. Further studies could associate these features to social behaviours and political strategies.

We also studied the different impact of detrending methods on the performance of warning signals. We showed that detrending time series with ARIMA models, appropriately calibrated for each country, increases the AUC score. This observation supports early theoretical works [50] and informs researchers about the importance of data processing methods to improve the performance of various indicators.

Finally, we analysed the potential limitations of the indicator system in other contexts, characterised by different dynamical features such as rapid increases of $R(t)$ and strong or non-additive noise. This emphasises that, for EWS to properly work, the real system must fulfill the discussed conditions underlying the theoretical modelling. These open problems highlight that knowledge of the type of collected data is imperative to avoid misleading judgements in response to time series trends: EWS as epidemiological constructs will only remain valuable and relevant when used and interpreted correctly [63].

We acknowledge the limitations of this study, which might be overcome when new and better curated data sets will become available. First, data quality could be a limiting factor, despite being representative of real world monitoring capacities. Reliable estimates of recovered and dead patients are necessary to guarantee the robustness of a proxy for active cases like $A$. Another potential data quality issue is that the official numbers of positives might still neglect undetected asymptomatic cases. The data set selection highlights the importance of monitoring and of high quality prevalence data (as already suggested in [14]). Secondly, our estimators come with uncertainties: the empirical $R(t)$ is an estimate of the true reproduction number that rests on the assumption of homogeneous dynamics, while the use of moving windows might yield odd behaviors of EWS [50] which can contribute to poor signals, in addition to rapid transition and non-white noise. Third, our definition of "ground truth" transition date is somewhat conservative, as we requested to have maximum probability of $R(t)$, the control parameter, to be greater than 1. In the real world, the appropriate detection threshold is conditional on the various costs of a late outbreak alert, and requires an assessment by public health authorities which could modify the estimated lead time. This aspect might also influence our interpretation of the "bifurcation delays": depending on the definition of the "ground truth", they might be less severe than what discussed before. Moreover, an alternative explanation for such observed delays might involve COVID-19 latency periods and reporting delays. When extra data are available, this aspect can be further elucidated using nowcasting methods [64]. Fourth, due to statistical uncertainties, a reliable estimation of the lead time—how much in advance a re-emergence can be predicted—was not entirely possible. Future studies will likely concentrate on this aspect, as early prediction would advance the current on-time detection.

This study uses a proxy $A$ for active cases (Eq 4) to compare theoretical results from various literature sources obtained from on prevalence data. To expand the testing of EWS on epidemic data, to compare them with more recent studies on incidence data [14, 22] and to avoid the potential biases associated with the proxy $A$ (see Discussion above), we also investigated the potential use of incidence data themselves. The related results and plots are displayed in Sec H in S1 Text. We observe consistency with the results here presented on prevalence data, but also some notable differences worth discussing. Firstly, we notice different noise distributions, diverging from Gaussian, that make the interpretation of the EWS performance more

challenging. Secondly, we observe a higher correlation between residuals from ARIMA and other detrending methods, possibly linked to weekly trends being mostly driven by testing routines and being equally smoothed. Finally, we observe an improved performance for the skewness over the $\mathcal{Y}$ test set, which contrasts with the results of [14], but is more in line with what suggested in [60]. We speculate that this might be related to the interplay of the approach to the transition and the noise distribution, but we limit ourselves to report the observation and to leave additional theoretical and computational analysis to further studies. Overall, such analysis still stresses that EWS performance is sensitive to the underlying modelling assumptions and, if not assessed carefully, could hinder our capability to extend them in uncertain contexts. In addition, the fact that performances over prevalence or incidence data are slightly different underline how much the approaches relying on EWS depend on the quantities that are measured and use for EWS calculation. Hence, from a practical point of view, looking at a measure or another might make a difference for the monitoring efficacy.

## Conclusion

In recognition that real epidemics might behave differently that what is commonly modelled, we nonetheless conclude that minimal dynamical models have the potential to predict relevant aspects of complex epidemics. While more detailed and complete multivariate models are being developed, macro-scale models based on complex systems theory can provide insights and indicators to detect epidemic re-emergence. On the one hand, our results begin supporting the theoretical literature findings and their basic assumptions; on the other, they warn against naive applications of summary statistics as EWS: if not correctly applied, they could return possibly misleading spurious signals. In addition, our findings call for future studies on forecasting techniques based on pattern recognition in different dynamical regimes. For instance, validated EWS could serve as basis for the feature selection of automated Machine Learning-based algorithms [18]. The dual synergy of theoretical predictions and empirical studies will continue to play a role in the field of epidemic control and will likely have a impact in informing public health decisions.

## Supporting information

**S1 Text. Supplementary text.** Sec A: Mathematical models and assumptions. Sec B: Data collection and curation. Sec C: Estimating R(t) with Bayesian inference using MCMC. Sec D in S1 Text: Determining the rate of approach for $R(t) \to 1$. Sec E: Evolution of EWS for all countries. Sec F: ARIMA detrending and corresponding global EWS. Sec G: Further investigation on AUC values. Sec H in S1 Text: Analysis of incidence data. Fig A: theoretical EWS for epidemic re-emergence. Fig B: Examples of discarded time series. Fig C: Curves of active cases for the considered countries. Fig D: Estimate of transition rate to $R$ critical value. Fig E: Evolution of the considered indicators for all countries. Fig F: Investigation of variance for countries in $\mathcal{N}$. Fig G: Analysis of the dynamical characteristics of the countries included in the data set, for incidence data. Fig H: Analysis of the residuals from the detrending methods. Fig I: Analysis of the variance in the Luxembourg setting. Fig J: ROC curves for each considered indicator. Fig K: Evolution of EWS far from the transition point. Table A: Additional information about the selected countries. Table B: ARIMA model parameter combinations. Table C: AUC scores for different indicators. Table D: ARIMA model parameter combinations over incidence data. (PDF)

## Acknowledgments

The authors thank the Research Luxembourg—COVID-19 Taskforce for mutual collaborations, and Professors P. Ashwin and A. Skupin for their useful feedback.

## Author Contributions

**Conceptualization:** Daniele Proverbio.

**Data curation:** Daniele Proverbio.

**Formal analysis:** Daniele Proverbio, Françoise Kemp.

**Funding acquisition:** Jorge Gonçalves.

**Investigation:** Daniele Proverbio, Françoise Kemp, Stefano Magni.

**Methodology:** Daniele Proverbio.

**Project administration:** Jorge Gonçalves.

**Software:** Daniele Proverbio, Françoise Kemp.

**Supervision:** Stefano Magni, Jorge Gonçalves.

**Visualization:** Daniele Proverbio.

**Writing – original draft:** Daniele Proverbio, Françoise Kemp.

**Writing – review & editing:** Daniele Proverbio, Françoise Kemp, Stefano Magni, Jorge Gonçalves.

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
