## [Decision Letter · Decision Letter 0]

28 Oct 2021

Dear Mr. Proverbio,

Thank you very much for submitting your manuscript "Performance of early warning signals for disease re-emergence: a case study on COVID-19 data" for consideration at PLOS Computational Biology.

As with all papers reviewed by the journal, your manuscript was reviewed by members of the editorial board and by several independent reviewers. In light of the reviews (below this email), we would like to invite the resubmission of a significantly-revised version that takes into account the reviewers' comments.

We cannot make any decision about publication until we have seen the revised manuscript and your response to the reviewers' comments. Your revised manuscript is also likely to be sent to reviewers for further evaluation.

Sincerely,

Benjamin Muir Althouse

Associate Editor

PLOS Computational Biology

Virginia Pitzer

Deputy Editor-in-Chief

PLOS Computational Biology

Reviewer's Responses to Questions

**Comments to the Authors:**

Reviewer #1: Review has been uploaded as an attachment.

Reviewer #2: See attachment

**Have the authors made all data and (if applicable) computational code underlying the findings in their manuscript fully available?**

Reviewer #1: Yes

Reviewer #2: Yes

PLOS authors have the option to publish the peer review history of their article (what does this mean?). If published, this will include your full peer review and any attached files.

Reviewer #1: No

Reviewer #2: No
---

## [Decision Letter · Decision Letter 1]

23 Feb 2022

Dear Mr. Proverbio,

We are pleased to inform you that your manuscript 'Performance of early warning signals for disease re-emergence: a case study on COVID-19 data' has been provisionally accepted for publication in PLOS Computational Biology.

Best regards,

Benjamin Althouse

Associate Editor

PLOS Computational Biology

Virginia Pitzer

Deputy Editor-in-Chief

PLOS Computational Biology

Reviewer's Responses to Questions

**Comments to the Authors:**

Reviewer #1: Proverbio et al. have dealt with my comments. I believe this is an interesting paper and well worth publishing.

Reviewer #2: In response to my comments the authors have heavily revised the manuscript text. The methods now read much more clearly. I am encouraged to see that the results are unchanged when incidence data, instead of the reconstructed prevalence data, is used to calculate the EWS. Personally, I would have preferred the analysis using incidence data to be in the main manuscript text (i.e. swapped with the analysis of prevalence data) as I feel biases in incidence data can be more readily identified. However, as the authors clearly state the respective issues associated with each data type (and the results agree), I feel that the quality of the paper is sufficient for publication.

**Have the authors made all data and (if applicable) computational code underlying the findings in their manuscript fully available?**

Reviewer #1: None

Reviewer #2: Yes

PLOS authors have the option to publish the peer review history of their article (what does this mean?). If published, this will include your full peer review and any attached files.

Reviewer #1: No

Reviewer #2: No

---

## [Editor Report · Acceptance letter]

25 Mar 2022

PCOMPBIOL-D-21-01654R1 

Performance of early warning signals for disease re-emergence: a case study on COVID-19 data

Dear Dr Proverbio,

I am pleased to inform you that your manuscript has been formally accepted for publication in PLOS Computational Biology. Your manuscript is now with our production department and you will be notified of the publication date in due course.

With kind regards,

Agnes Pap
